# A native interactor scaffolds and stabilizes toxic ATAXIN-1 oligomers in SCA1

Cristian A Lasagna-Reeves[1,2], Maxime WC Rousseaux[1,2], Marcos J Guerrero-Muñoz[3], Jeehye Park[1,2], Paymaan Jafar-Nejad[1,2], Ronald Richman[4,2], Nan Lu[1,2], Urmi Sengupta[3], Alexandra Litvinchuk[1], Harry T Orr[5], Rakez Kayed[3], Huda Y Zoghbi[4,2]*

[1]Department of Molecular and Human Genetics, Baylor College of Medicine, Houston, United States; [2]Jan and Dan Duncan Neurological Research Institute, Texas Children's Hospital, Houston, United States; [3]Department of Neurology, University of Texas Medical Branch, Galveston, United States; [4]Department of Molecular and Human Genetics, Howard Hughes Medical Institute, Baylor College of Medicine, Houston, United States; [5]Institute for Translational Neuroscience, University of Minnesota, Minnesota, United States

**Abstract** Recent studies indicate that soluble oligomers drive pathogenesis in several neurodegenerative proteinopathies, including Alzheimer and Parkinson disease. Curiously, the same conformational antibody recognizes different disease-related oligomers, despite the variations in clinical presentation and brain regions affected, suggesting that the oligomer structure might be responsible for toxicity. We investigated whether polyglutamine-expanded ATAXIN-1, the protein that underlies spinocerebellar ataxia type 1, forms toxic oligomers and, if so, what underlies their toxicity. We found that mutant ATXN1 does form oligomers and that oligomer levels correlate with disease progression in the $Atxn1^{154Q/+}$ mice. Moreover, oligomeric toxicity, stabilization and seeding require interaction with Capicua, which is expressed at greater ratios with respect to ATXN1 in the cerebellum than in less vulnerable brain regions. Thus, specific interactors, not merely oligomeric structure, drive pathogenesis and contribute to regional vulnerability. Identifying interactors that stabilize toxic oligomeric complexes could answer longstanding questions about the pathogenesis of other proteinopathies.

*For correspondence: hzoghbi@bcm.edu

## Introduction

The family of neurodegenerative diseases known as proteinopathies are characterized neuro-pathologically by deposits of insoluble proteins. In Alzheimer disease (AD), for example, there are extracellular depositions of amyloid-$\beta$ (A$\beta$) plaques and intracellular accumulations of neurofibrillary tangles of tau (*Glenner and Wong, 1984*; *Grundke-Iqbal et al., 1986*). Similarly, in Parkinson disease (PD), $\alpha$-synuclein forms cytoplasmic inclusions known as Lewy bodies (*Lashuel et al., 2013*). Although AD, PD, and other proteinopathies such as Huntington disease (HD) and the prion disorders are associated with different proteins possessing various structures, functions, and affecting different brain regions, they all involve the accumulation of $\beta$-sheet-rich entities, which suggests some commonality of pathogenic mechanism (*Glabe and Kayed, 2006*; *Knowles et al., 2014*).

Like HD, Spinocerebellar ataxia type 1 (SCA1) is a polyglutamine disease that belongs to this larger class of proteinopathies. SCA1 is caused by the expansion of a CAG repeat that encodes for glutamine (Q) in ATAXIN-1 (ATXN1), a protein that is expressed throughout the brain. The Atxn1 knockin mouse model, which bears a 154 CAG repeat knocked into the murine Atxn1 locus, faithfully reproduces the SCA1 phenotype: progressive motor incoordination due to cerebellar degeneration, cognitive deficits, premature death, and degradation-resistant deposits (nuclear inclusions, or NIs) that contain mutant

**eLife digest** Spinocerebellar ataxia type 1 (SCA1) is a progressive neurodegenerative disease in which damage to the brain regions that control movement results in the gradual loss of coordination and motor skills. The disease is a caused by a mutation in the gene that codes for a protein called ATAXIN-1. In healthy individuals this protein contains up to 39 copies of an amino acid called glutamine. However, the mutant gene can encode for 40 or more copies of glutamine, which results in a longer-than-usual ATAXIN-1 protein with toxic properties.

Within the brain, some of the toxic ATAXIN-1 proteins form insoluble deposits, while the rest remain soluble. At first it was assumed that the insoluble deposits were responsible for the neurodegeneration seen in SCA1. However, closer examination revealed that these deposits form mainly in brain regions that do not degenerate, which suggests that they might instead have a protective role. This is consistent with evidence from research into other brain disorders, including Alzheimer's disease, which suggests that the soluble form of the toxic proteins might be causing these diseases.

Lasagna-Reeves et al. now provide the first direct evidence that the soluble form of the toxic ATAXIN-1 proteins are indeed harmful in a mouse model of SCA1. Experiments reveal that these soluble proteins accumulate in the brain regions that undergo degeneration in SCA1, such as the cerebellum, but not in those regions that remain intact. Moreover, the motor skills and coordination of the mice get worse as the level of soluble toxic ATAXIN-1 increases.

Lasagna-Reeves et al. go on to show that a protein called capicua stabilizes the toxic ATAXIN-1 proteins, which keeps them from forming insoluble deposits. Since capicua is particularly abundant in the cerebellum, this could explain the high levels of toxic ATAXIN-1 in this region and why it is vulnerable to degeneration.

Future experiments are needed to investigate whether proteins equivalent to capicua might play a similar role in stabilizing toxic proteins in Alzheimer's, Parkinson's and Huntington's diseases, and whether preventing this stabilization could have therapeutic potential.

ATXN1 (*Watase et al., 2002*). As with the stable fibrillar deposits first observed in AD over a hundred years ago, the prominence of these NIs led initially to the postulate that this material is the causative agent of disease (*Chiti and Dobson, 2006*). Yet the NIs develop primarily in neurons that escape degeneration, not in the cerebellar Purkinje cells (PCs), which are the first to succumb to SCA1 pathology (*Watase et al., 2002*). This curious observation led to the proposal that the ATXN1-containing NIs are not themselves toxic but rather might serve a protective role by sequestering the mutant protein (*Cummings et al., 1998*, *1999*).

Recent findings suggest a refinement to this hypothesis: it may be that the primary drivers of toxicity are metastable non-fibrillar species known as soluble oligomers (*Glabe, 2008*; *Benilova et al., 2012*; *Krishnan et al., 2012*). Although toxic oligomers have been identified in HD models and their modulation relates to beneficial outcomes (*Legleiter et al., 2010*; *Sontag et al., 2012*) their specific role in disease progression in vivo remains unstudied. Furthermore, there are not studies regarding the role of binding partners of the disease-related proteins in the oligomerization process. The inverse correlation between NIs and neuronal integrity in SCA1, however, lends appeal to the hypothesis that soluble oligomers, rather than fibrils per se, drive neurodegeneration in SCA1.

In this study we sought to determine if and how oligomeric forms of polyQ ATXN1 might contribute to the SCA1 disease state. We report the discovery of polyQ ATXN1 oligomers in the *Atxn1* knockin mouse and demonstrate that these oligomers do indeed correlate with disease pathogenesis and motor dysfunction. We also show that polyQ ATXN1 oligomers seed the formation of new oligomers and demonstrate that Capicua (CIC), a key native binding partner of ATXN1, plays a pivotal role in the stabilization and regional toxicity of these oligomeric species.

## Results

### ATXN1 oligomers are associated with neurodegeneration in SCA1

In the absence of high-resolution structural data for oligomers, conformation-dependent antibodies can be used to distinguish between different types of amyloid structures by recognizing epitopes that

are associated with specific aggregation states, independent of their amino acid sequences (*Kayed et al., 2003*, *2010*). We used the conformational monoclonal anti-oligomer antibody F11G3 to detect ATXN1 oligomers in the *Atxn1*$^{154Q/+}$ knockin mouse model. This antibody has been extensively characterized and compared to other anti-oligomer antibodies previously developed using similar methods (*Guerrero-Munoz et al., 2014a*, *2014b*). Oligomers were apparent in cerebellar extracts of *Atxn1*$^{154Q/+}$ but not in wild-type or *Atxn1*$^{-/-}$ mice (*Figure 1A*). To confirm the anti-oligomeric nature of F11G3, we pre-incubated the antibody with different types of oligomers prior to performing IF in brain sections from *Atxn1*$^{154Q/+}$ mice. The results verified that F11G3 is indeed highly specific to an oligomeric conformation rather than an amino acidic sequence (*Figure 1—figure supplement 1*). Immunofluorescence (IF) against both ATXN1 and oligomers revealed substantial co-localization in the cerebellum (*Figure 1B*). Immunoprecipitation of oligomers from the *Atxn1*$^{154Q/+}$ cerebellum confirmed that these metastable entities are formed by ATXN1 (*Figure 1C*). Atomic force microscopy (AFM) images show that these oligomers have an average height of 6.8 +/− 3.4 nm (*Figure 1D*).

If oligomeric ATXN1 does indeed drive pathology, it should be most abundant in cerebellar Purkinje cells, the cells most vulnerable to SCA1. Staining of PCs at various stages of neurodegeneration in the *Atxn1*$^{154Q/+}$ mice revealed high levels of oligomers in the neurites prior to their degeneration; in later stages, oligomers accumulated in the soma (*Figure 1E*). When we explored the regional localization of oligomeric forms of ATXN1, we found them associated with neurons that are prone to degeneration (e.g., PCs) but not those that are spared by the disease (e.g., cortical neurons; *Figure 1F* and *Figure 1—figure supplement 2*). The presence of ATXN1 oligomers in more susceptible neurons suggests that these soluble aggregates are important in the neuropathology of SCA1.

## ATXN1 amyloid oligomers correlate with disease progression

We took advantage of the fact that the knockin mouse model reproduces the progressive phenotype of SCA1 (*Zoghbi and Orr, 2009*) to investigate the relationship between oligomer levels and degree of motor impairment in the *Atxn1*$^{154Q/+}$ mice. Motor incoordination in the rotarod assay becomes apparent in these mice as early as 5 weeks of age and worsens throughout their life span (*Watase et al., 2002*). Accordingly, we tested the levels of ATXN1 oligomers in these mice at an early stage of disease (8 weeks of age), a middle stage (18 weeks), and a late stage (28 weeks). There was considerable variability among animals of the same age in both motor performance and oligomer abundance, but the levels of different molecular weight oligomers in the cerebellum clearly increased as the mice aged (*Figure 2A*). The differences between individual mice, however, allowed us to search for relationships between motor impairment and different ATXN1 oligomers. We tested each mouse on the rotarod and observed a consistently strong inverse correlation between latency to fall and the level of ATXN1 oligomers (*Figure 2B*), with oligomers over 245 KDa showing the greatest inverse correlation (R = 0.89). Importantly, no significant changes in monomeric ATXN1 levels (2Q or 154Q) were observed in these mice (*Figure 2—figure supplement 1*). Thus, oligomer levels correlated with disease progression as assessed by motor incoordination.

To determine the native size of soluble ATXN1 oligomers under non-denaturing conditions to insure that they were not artificially generated during SDS-PAGE, we performed non-denaturing size-exclusion chromatography (SEC) of cerebellar samples and subjected these to SDS-Tris-Glycine gels (*Figure 2C*). Evaluation of individual fractions by western blot with anti-ATXN1 antibody revealed WT and polyQ monomeric ATXN1 in different fractions, as previously shown (*Figure 2C*, right panel) (*Lam et al., 2006*). When we analyzed the SEC fractions with the anti-oligomer antibody, we observed oligomeric species mainly between fractions 10 and 12 (ranging from 500 to 1000 KDa in size; *Figure 2C*, left panel). Anti-ATXN1 antibody confirmed that the oligomers in these fractions do in fact contain ATXN1 (*Figure 2C*, right panel); the range of sizes of these oligomers is thus rather broad. Fraction 11 (~667 KDa) contained a peak in the signal of oligomers >245 KDa in weight, as well as the presence of lower molecular weight oligomeric species (~180, ~135, ~110 and ~98 KDa). AFM revealed that those fractions detected by the anti-oligomer antibody (particularly fractions 11 and 12) were indeed oligomeric, whereas the higher molecular weight fractions were fibrillar in nature (fraction 7, ~4000KDa; *Figure 2D*). A hydrophobic interaction column, which separates proteins according to hydrophobic domain exposure, showed that fraction 11 is highly hydrophobic, as is characteristic of many oligomers (*Campioni et al., 2010*) (*Figure 2—figure supplement 2*). To establish how stable these oligomers are, SEC fractions were incubated at 37°C for up to 48 hr followed by immunoblot analysis using F11G3. F11G3 showed a strong affinity for fractions

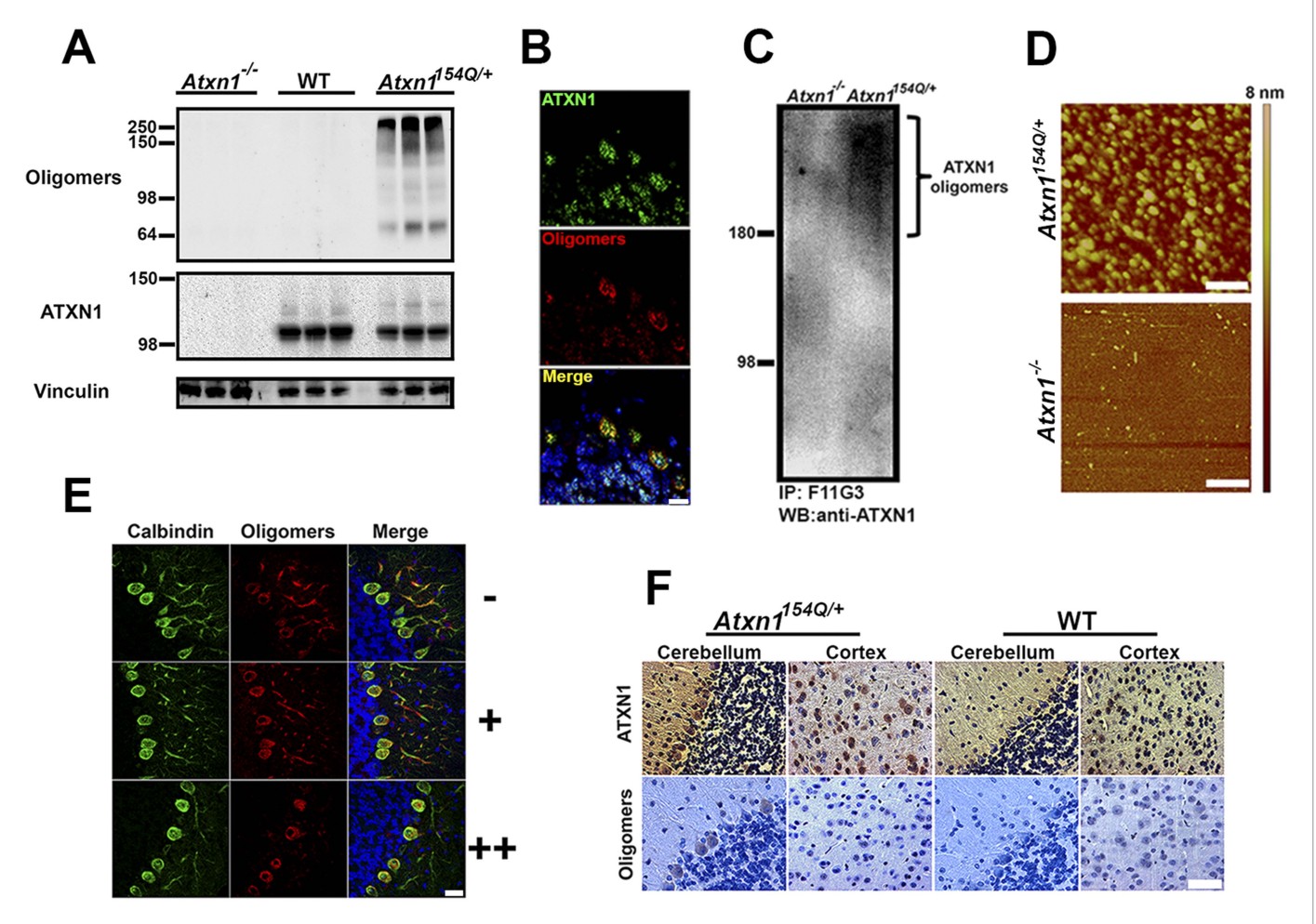

**Figure 1**. ATXN1 oligomers are located in areas prone to SCA1 degeneration. (**A**) Western blot analysis of soluble fractions from cerebella shows the existence of amyloid oligomers exclusively in $Atxn1^{154Q/+}$ mice model but in neither wild-type nor $Atxn1^{-/-}$ controls. (**B**) Immunofluorescence studies of $Atxn1^{154Q/+}$ brain sections using anti-Ataxin-1 (green) and F11G3 (red) confirms the ATXN1 identity of the amyloid oligomers. Scale bar = 15 µm. (**C**) Western blot using anti-ATXN1 antibody showed that the isolated oligomers (IP with F11G3), were indeed ATXN1 oligomers. These ATXN1 oligomers were IP'd exclusively from $Atxn1^{154Q/+}$ mouse cerebellum and not from $Atxn1^{-/-}$ controls. (**D**) AFM analysis showing brain-derived ATXN1 oligomers IP from $Atxn1^{154Q/+}$ mouse using F11G3. Scale bar 200 nm. (**E**) Double staining using the Purkinje cell (PC) marker, calbindin (green) and F11G3 (red) revealed that ATXN1 oligomers accumulate in PC dendrites before dendritic degeneration is observed (top panel). With progression of dendritic degeneration, ATXN1 oligomers tend to accumulate in both the cytoplasm and nucleus (middle and bottom panel). Scale bar 30 µm. No degeneration is represented by (−), a low level of degeneration is represented by (+), and advanced degeneration by (++). Mice analyzed were 28 weeks old. (**F**) Histological staining for ATXN1 (top panel) and oligomers (F11G3, bottom panel) in cerebellum and cortex of $Atxn1^{154Q/+}$ (left panels) and WT mice (right panels). Scale bar 50 µm.

The following figure supplements are available for figure 1:

**Figure supplement 1**. ATXN1 oligomers in SCA1 mouse.

**Figure supplement 2**. ATXN1 oligomers levels are higher in the cerebellum than in the cortex.

11 and 12 even after 48 hr of incubation, suggesting that these oligomers are highly stable (*Figure 2—figure supplement 3*).

To determine whether the ATXN1 oligomeric complex can induce cellular toxicity, we treated cultured cerebellar granular neurons (CGNs) with SEC fractions from $Atxn1^{154Q/+}$ cerebellar samples. Cell viability measurement revealed fraction 11 to be highly toxic (*Figure 2E*). This toxicity was blocked when fraction 11 was pre-incubated with the anti-oligomer antibody F11G3 or anti-ATXN1 antibody.

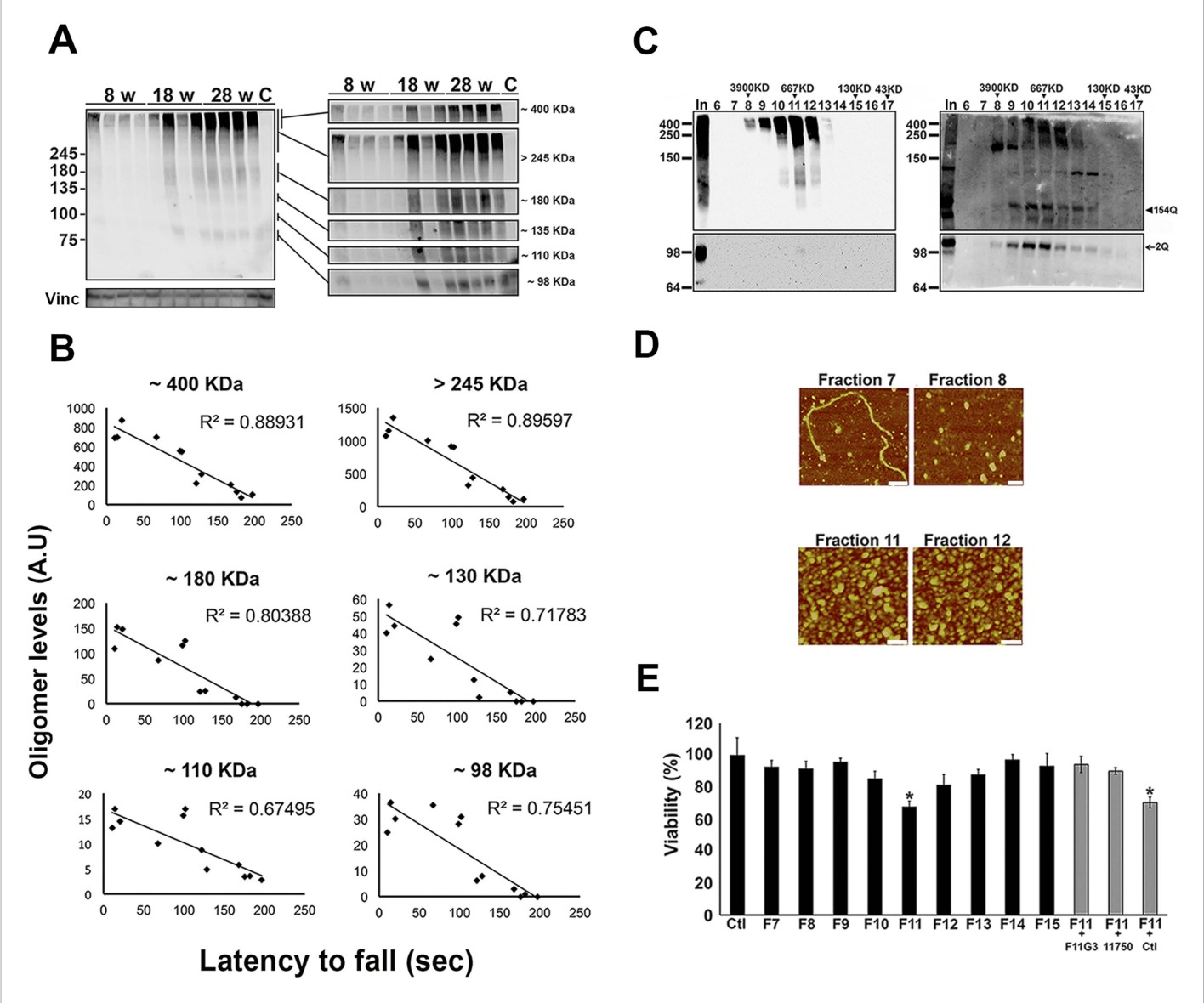

**Figure 2**. ATXN1 oligomers form high molecular weight complexes that correlate with motor impairment. (**A**) Western blot of cerebellar lysates of 8-, 18- and 28-week old *Atxn1^{154Q/+}* mice using F11G3. Right panels indicate cropped regions of the blot with exposure adjusted to permit comparative quantification among mice of different ages. (**B**) Correlation plots between the different molecular weight oligomers (*Figure 1A*) and the latency to fall in the rotarod assay in *Atxn1^{154Q/+}* mice. Each black dot corresponds to one *Atxn1^{154Q/+}* mouse from 1A. (**C**) Size exclusion chromatography (SEC) from *Atxn1^{154Q/+}* mouse cerebellum probed for oligomers (F11G3, left panel) and ATXN1 (11750, right panel). Note that top panels reveal a higher exposure than the bottom panels. *In* corresponded to Input before fractionation. (**D**) Representative atomic force microscopy (AFM) pictures from fractions 7, 8, 11 and 12 (Scale bar 140 nm). (**E**) MTT (3-(4,5-Dimethylthiazol-2-yl)-2,5-Diphenyltetrazolium Bromide) assay performed on cerebellar granule neurons. Cells grown 7 days in vitro (DIV) were treated with different fractions (labeled below the *x*-axis) and viability was measured 18 hr following treatment (black bars). Gray bars denote viability following incubation of fraction 11 previously co-incubated with the indicated antibodies. *p < 0.05.

The following figure supplements are available for figure 2:

**Figure supplement 1**. Correlation between the levels of 154Q or 2Q ATXN1 and the latency to fall from the rotarod.

**Figure supplement 2**. ATXN1 oligomers characterization by hydrophobic interaction column.

**Figure supplement 3**. ATXN1 oligomers from *Atxn1^{154Q/+}* are highly stable.

**Figure supplement 4**. ATXN1 oligomeric complex induce cellular toxicity.

The cellular damage was thus produced by an ATXN1-containing oligomeric complex present in this SEC fraction. IF staining of CGNs with the neuronal marker Tuj1 revealed that the ATXN1 oligomer complex from fraction 11-induced neurodegeneration, as evidenced by neuritic swelling and beading (*Figure 2—figure supplement 4*).

## ATXN1 oligomers seed the formation of endogenous ATXN1 oligomers in cell culture

Neuropathological analysis of the *Atxn1*[154Q/+] mice revealed that ATXN1 oligomers are not evenly distributed throughout the cerebellum but instead are restricted to focal sub-populations of PCs. It is notable that this focal distribution coincided with cellular toxicity: cells with no detectable oligomers looked healthy, displaying many calbindin-positive PCs with extensive projections, while areas with abundant oligomers showed more diffuse calbindin staining and short dendrites (*Figure 3A*). Given that, in AD and PD, proteins such as Aβ, tau or α-synuclein appear to 'seed' the accumulation of other neurotoxic proteins and promote neurodegeneration (*Clavaguera et al., 2009*; *Langer et al., 2011*; *Lasagna-Reeves et al., 2012*; *Rey et al., 2013*), we sought to determine whether a similar mechanism operates in SCA1.

To evaluate the potential seeding activity of ATXN1 oligomer complexes, we used a previously described cellular model that expresses ATXN1(82Q) fused to monomeric red fluorescent protein (mRFP) (*Park et al., 2013*). These cells form nuclear inclusions, but their capacity to form oligomers had never been evaluated. We first established that these cells do form oligomers (*Figure 3—figure supplement 1*). As a negative control, we used cells that express mRFP-ATXN1(82Q,S776A), a mutant form of ATXN1 that cannot be phosphorylated at Ser776 and is not toxic (*Emamian et al., 2003*; *Jorgensen et al., 2009*; *Park et al., 2013*). The anti-oligomer antibody detected the smaller inclusion bodies (350–900 nm) but did not recognize the larger inclusions (>900 nm), suggesting that these larger inclusions are more complex and composed of higher order aggregates, likely to be fibrillar in nature.

We next tested whether ATXN1 oligomer complexes from *Atxn1*[154Q/+] mouse cerebellum seed the formation of new oligomers in this cell model. We added cerebellar fractions (see *Figure 2C*) to the cells and quantified the percentage of mRFP-ATXN1(82Q) cells with oligomeric or fibrillar inclusions, based on size (350–900 nm or >900 nm, respectively) and whether or not the inclusions were detected by F11G3. The cells exposed to either fraction 11 or 12 developed the largest proportion of oligomeric inclusions (55% and 34%, respectively) in comparison with cells incubated with other fractions or the control group (18%) (*Figure 3B*). None of the treated groups showed an increase in fibrillar inclusions. This suggests that ATXN1 oligomeric complexes seed the formation of new oligomers but do not promote the formation of ATXN1 fibrillar material. No fibrillar or oligomeric inclusions were observed in any of the mRFP-ATXN1(82Q, S776A) cells, regardless of the treatment (data not shown). ATXN1 oligomeric complexes may thus act as a seed, but the cell needs to express an entity that is prone to aggregate.

To confirm that the seeding effect was in fact produced by ATXN1 oligomers, Daoy mRFP-ATXN1(82Q) cells treated with Fraction 11 were immunostained with either anti-oligomer antibody (*Figure 3C*) or an anti-ATXN1 antibody (*Figure 3D*). The anti-oligomer staining (green fluorescence) revealed several oligomeric complexes that lacked an mRFP signal, suggesting that exogenously added oligomers were internalized by the cell (*Figure 3C*). ATXN1 staining revealed that cells also internalized exogenously added ATXN1 species (*Figure 3D*). We next treated Daoy mRFP-ATXN1(82Q) cells with fraction 11 pre-incubated with the anti-oligomer antibody, the anti-ATXN1 antibody, or a control antibody. Cells treated with fraction 11 pre-incubated with the anti-oligomer or the anti-ATXN1 antibody blocked oligomer formation (*Figure 3E*). It is therefore the ATXN1 oligomers present in fraction 11 that are responsible for inducing the formation of new oligomeric inclusions in mRFP-ATXN1(82Q) cells.

To determine the mechanism through which the anti-oligomer antibody blocks the seeding effect of ATXN1 oligomers, we treated Daoy mRFP-ATXN1(82Q) cells with fraction 11 pre-incubated with the anti-oligomer antibody and performed IF using only a secondary antibody labeled with alexa 488. Fluorescent images show that the anti-oligomer antibody was not internalized by the cell but bound to the ATXN1 oligomers in the extracellular space. The antibody thus appears to block the internalization of exogenous ATXN1 oligomers rather than interacting directly with intracellular oligomeric species. The presence of anti-oligomer

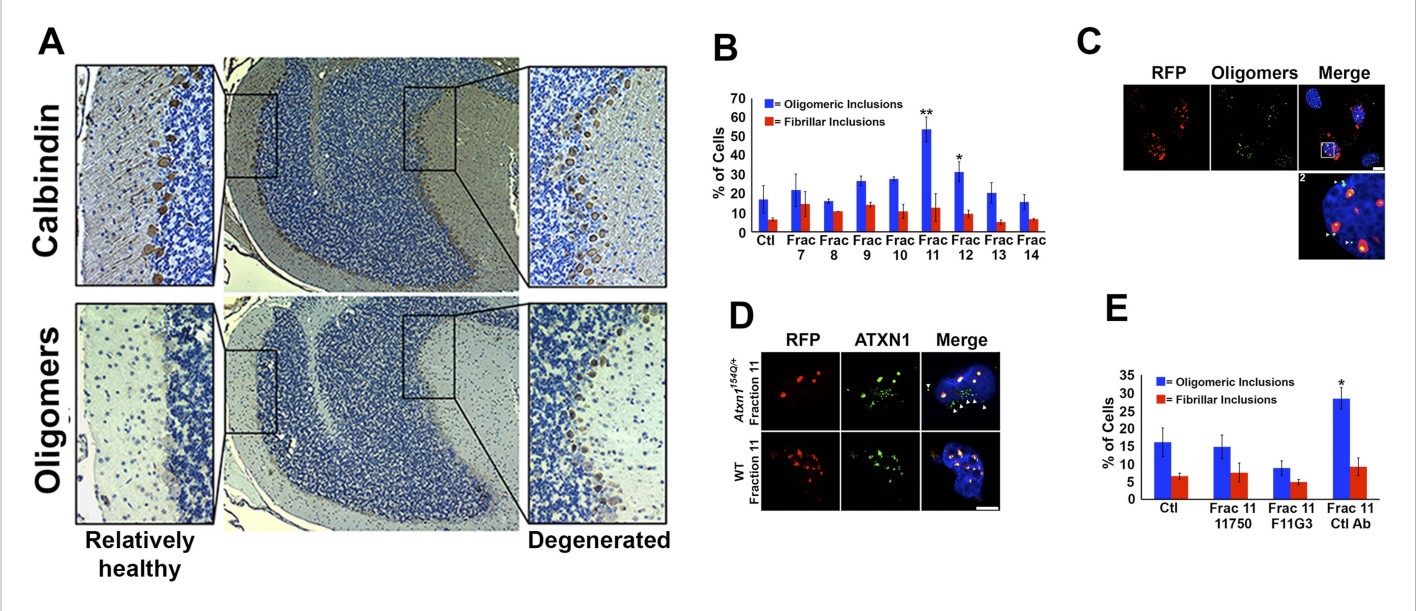

**Figure 3**. ATXN1 oligomer complexes from *Atxn1*[154Q/+] mouse cerebellum induce the formation of new ATXN1 oligomers. (**A**) Histological analysis in *Atxn1*[154Q/+] brain sections reveals the regional presence of oligomers in PCs stained with calbindin (top panel) and oligomers stained with F11G3 (bottom panel) in degenerating PCs (bottom right panel) but not surviving PCs (bottom left panel). (**B**) Cell-based seeding assay: mRFP-ATXN1(82Q) cells were incubated with the indicated fractions (Frac 7–14) and the oligomeric and fibrillar inclusions were quantified. *p < 0.05. **p < 0.01. (**C**) Immunostaining for oligomers (F11G3, green) of mRFP-ATXN1(82Q) cells following treatment with fraction 11. Arrowheads denote internalized oligomers. Scale bar 15 µm. (**D**) Immunostaining for ATXN1 (11750, green signal) of mRFP-ATXN1(82Q) after treatment with fraction 11 from *Atxn1*[154Q/+] cerebellum (top row) demonstrated that exogenous ATXN1 (green signal) is internalized into the cells. Cells treated with fractions 11 from wild-type mouse cerebellum (bottom row) did not show internalized ATXN1 entities. Scale bar 15 µm. (**E**) Quantification of the percentage of mRFP-ATXN1(82Q) cells with oligomeric (blue bars) or fibrillar (red bars) inclusions following treatment of fraction 11 pre-incubated with the indicated antibodies *p < 0.05.

The following figure supplements are available for figure 3:

**Figure supplement 1**. ATXN1 oligomers are detected in mRFP-ATXN1(82Q) cells.

**Figure supplement 2**. Anti-oligomer antibody F11G3 does not get internalized into the cells.

antibody was confirmed by western blot analysis of media from cells treated with only secondary antibody (*Figure 3—figure supplement 2*).

## CIC interacts with ATXN1 oligomers stabilizing the complex in its oligomeric conformation

In SCA1, both wild-type and expanded ATXN1 share many binding partners (*Lim et al., 2006*). Among these, the transcriptional repressor CIC is of particular interest, because it is the only known binding partner whose protein levels are significantly reduced in *Atxn1*[−/−] mice (*Lee et al., 2011*); its stability is thus dependent on being complexed with ATXN1 (*Lam et al., 2006*). Moreover, our lab has shown that polyQ ATXN1 exerts toxicity through its native CIC-containing complex rather than through aberrant interactions with novel proteins (*Lam et al., 2006*), and that reducing CIC levels by half rescues many SCA1-like phenotypes in *Atxn1*[154Q/+] mice (*Fryer et al., 2011*).

When we performed SEC on cerebellar extracts from *Atxn1*[154Q/+] mice, we detected CIC in the same fractions enriched with ATXN1 oligomer complexes. These fractions lost their seeding capacity when pre-incubated with an anti-CIC antibody (*Figure 4—figure supplement 1*). Based on these observations, we investigated the role of CIC in the formation and stability of ATXN1 oligomer complexes. We generated *Atxn1*[154Q/+];*Cic*[L+/-] mice that have a ~50% reduction of both isoforms of CIC (Cic-L and Cic-S) (*Fryer et al., 2011*) and observed a significantly lower number of PCs positive for oligomers in these mice (*Figure 4A,B*). This result was confirmed by western blot analyses on whole cerebellar extracts from these animals (*Figure 4C,D*).

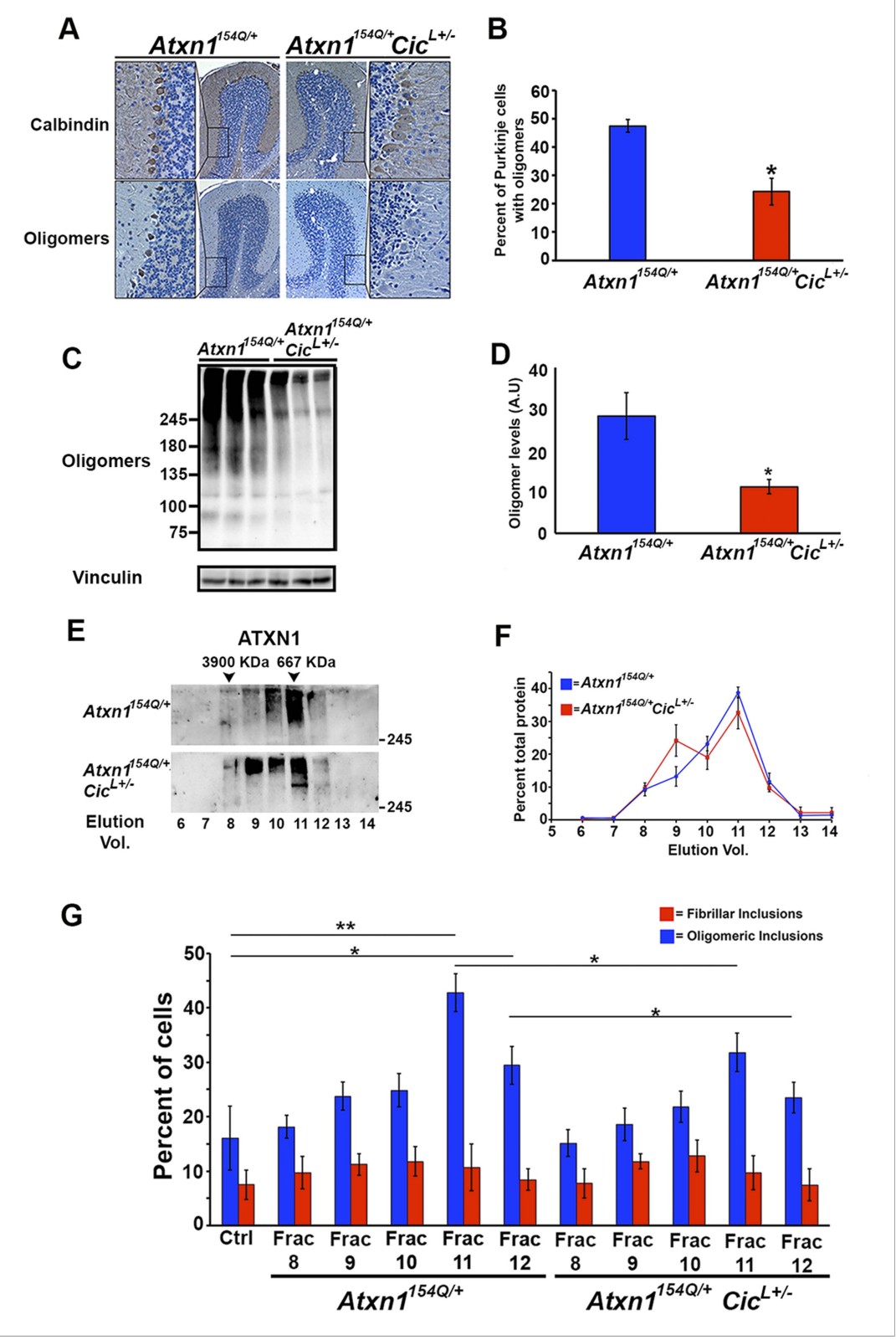

**Figure 4**. Reduction of CIC levels decreases the levels of ATXN1 oligomer complexes in vivo. (**A–B**) Pathological analysis showed fewer PCs from *Atxn1*[154Q/+];*Cic*[L+/-] mice loaded with oligomers than PCs from *Atxn1*[154Q/+] mice. PCs were stained with calbindin antibody and oligomers with F11G3. *p < 0.05. (**C–D**) Western blot revealed lower

*Figure 4. Continued*

levels of oligomers in the cerebellum of *Atxn1^154Q/+^;Cic^L+/-^* mice than in *Atxn1^154Q/+^* mice. Oligomers were measured using F11G3. **p < 0.01. (**E–F**) Size Exclusion Chromatography (SEC) showed that ATXN1 complexes in cerebellar samples from *Atxn1^154Q/+^;Cic^L+/-^* mice shift to a higher molecular weight fraction than those in *Atxn1^154Q/+^* cerebella. (**G**) Cell-based seeding assay demonstrated that the absence of one *Cic* allele in *Atxn1^154Q/+^;Cic^L+/-^* mice reduced the formation of new ATXN1 oligomers in Daoy mRFP-ATXN1(82Q). Fraction 11 and 12 ATXN1 oligomer complexes from control *Atxn1^154Q/+^* mouse cerebella seeded the formation of new mRFP-ATXN1(82Q). *p < 0.05, **p < 0.01.

The following figure supplements are available for figure 4:

**Figure supplement 1**. CIC is part of the ATXN1 oligomeric complex.

**Figure supplement 2**. Higher molecular weight complexes in *Atxn1^154Q/+^;Cic^L+/-^* do not present an oligomeric conformation.

**Figure supplement 3**. High molecular weight complexes characterization by filter retardation assay (FRA).

To our surprise, SEC on cerebellar extracts from *Atxn1^154Q/+^;Cic^L+/-^* and *Atxn1^154Q/+^* mice revealed that decreasing the levels of CIC induced a shift of the ATXN1 complexes to a higher molecular weight fraction detected by ATXN1 antibody (*Figure 4E,F*). Furthermore, these higher molecular weight complexes are not oligomeric: anti-oligomer antibody did not reveal any shift to the higher molecular weight fractions upon loss of Cic (*Figure 4—figure supplement 2*). To confirm that the high molecular weight complex present only in *Atxn1^154Q/+^;Cic^L+/-^* mice adopted a more organized fibrillar aggregate conformation, we performed Filter Retardation Assay (FRA) using the SEC fractions from *Atxn1^154Q/+^;Cic^L+/-^* and *Atxn1^154Q/+^* mice. Probing the FRA membrane with anti-ATXN1 antibody revealed the presence of ATXN1 SDS-insoluble aggregates in fractions 7 and 8 in *Atxn1^154Q/+^* mice. In the case of the *Atxn1^154Q/+^;Cic^L+/-^* mice, ATXN1 SDS-insoluble aggregates were also present in fraction 9 (*Figure 4—figure supplement 3*), confirming that in the context of reduced CIC levels, ATXN1 tends to form more organized fibrillar aggregates. We also probed FRA membranes with OC, a conformational antibody that specifically recognizes fibrillar oligomers and fibrils (*Kayed et al., 2007*), and found fractions 7 and 8 in *Atxn1^154Q/+^* mice, and fractions 7, 8, and 9 in *Atxn1^154Q/+^;Cic^L+/-^* mice, positive for OC (*Figure 4—figure supplement 3*). ATXN1 thus tends to form fibrillar oligomers, amorphous aggregates and/or fibrils when CIC levels are reduced. We detected no oligomers using F11G3 in the FRA membranes; ATXN1 oligomers detected by F11G3 are therefore soluble in 2% SDS (*Figure 4—figure supplement 3*) and able to escape detection by FRA. To determine whether the ATXN1 complex maintains its seeding capacity in the absence of one *Cic* allele, we performed the cell-based seeding assay using mRFP-ATXN1(82Q) cells treated with SEC fractions from either *Atxn1^154Q/+^* or *Atxn1^154Q/+^;Cic^L+/-^* cerebellum and quantified the percentage of cells containing fibrillar or oligomeric inclusions. Fractions 11 and 12 from the *Atxn1^154Q/+^;Cic^L+/-^* mice were less proficient in inducing the formation of new oligomeric inclusions than those from the *Atxn1^154Q/+^* mice (*Figure 4G*).

To determine whether CIC plays a key role in ATXN1 oligomerization, we co-transfected Hela cells with a consistent amount of ATXN1(82Q) and varying quantities of CIC. As a negative control, we utilized CIC with the W37A mutation which hinders its interaction with ATXN1 (*Kim et al., 2013*). Increasing amounts of CIC led to increasing levels of oligomers as measured with F11G3 (*Figure 5A,B*), but increasing amounts of mutant CIC/W37A produced no changes in the levels of ATXN1 oligomers in the cells. To determine whether polyQ-length plays a role in CIC's stabilization of ATXN1 oligomers, we repeated the same co-transfection experiments in HeLa cells as described for ATXN1(82Q), this time utilizing nonpathogenic wild-type ATXN1(30Q). When cells were co-transfected with ATXN1 (30Q) and increasing amounts of CIC, oligomers were almost undetectable by western blot using F11G3 (*Figure 5C*). As expected, increasing amounts of mutant CIC/W37A produced no oligomeric signal. To compare the relative amounts of oligomers in the ATXN1(82Q) and ATXN1 (30Q) cells, we performed direct ELISA using F11G3 and normalized the values by the total amount of ATXN1 measured using anti-ATXN1 antibody. The ELISA quantification confirmed that increasing amounts of CIC corresponded to increasing levels of oligomers in ATXN1(82Q) cells, whereas no changes in oligomer levels were observed when cells were treated with any amount of mutant CIC/W37A. Cells

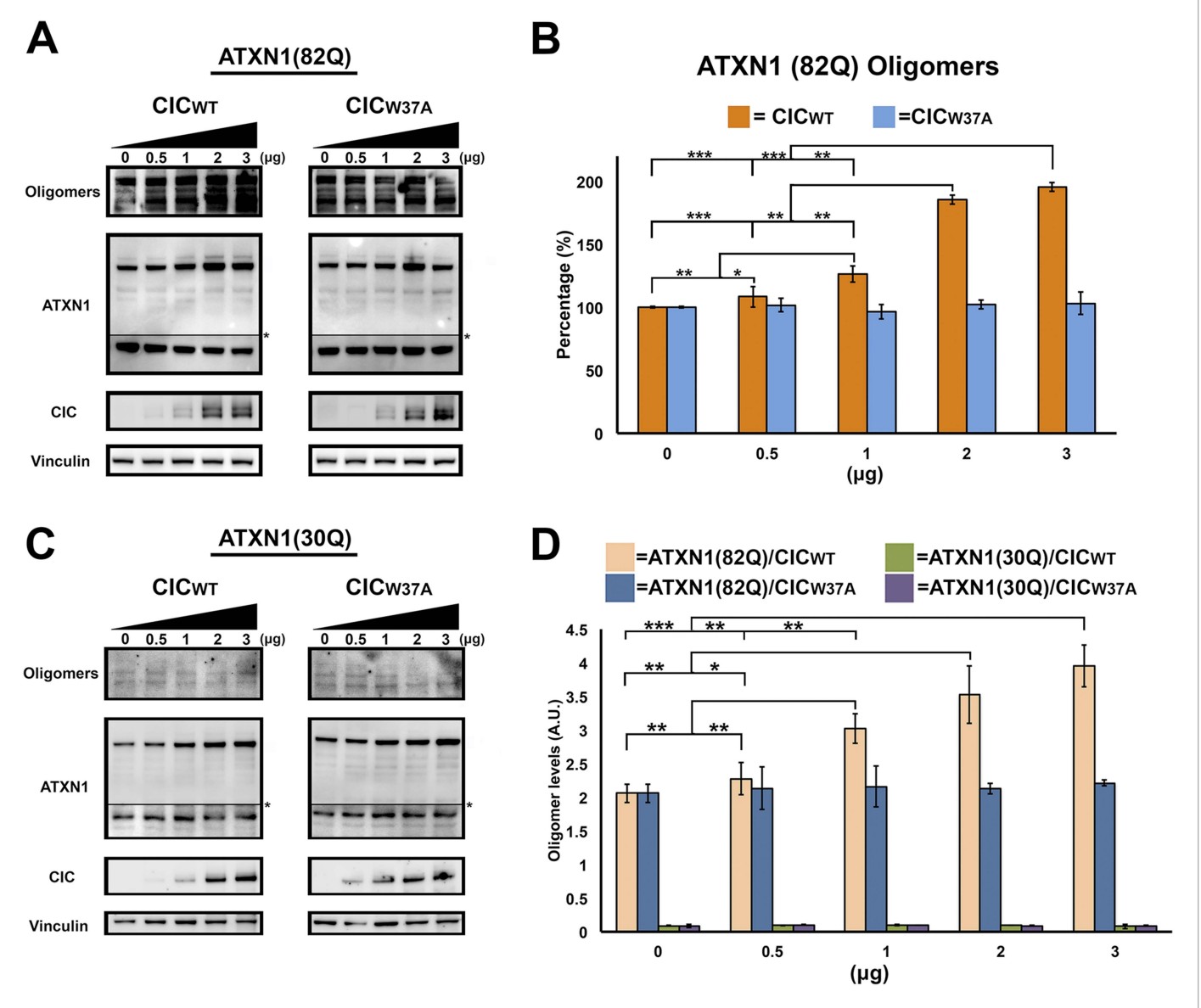

**Figure 5**. ATXN1 requires CIC binding to maintain oligomer conformation. (**A**) Western blot of Hela cells co-transfected with ATXN1(82Q) and increasing concentrations (from 0.5 to 3 μg) of wild type CIC (left panel) or increasing concentrations of mutant CIC/W37A (right panel). Membranes were probed with F11G3 to determine the levels of oligomers. * indicated change in exposure of the membrane. (**B**) Quantification of oligomers from Western blot analysis (**A**), demonstrated that increased oligomer levels correlated with increases of transfected wild-type CIC. Mutant CIC/W37A produced no effect on the amount of oligomers detected by Western blot. n = 4. *p < 0.05, **p < 0.01 and ***p < 0.001. (**C**) Western blot of Hela cells co-transfected with ATXN1(30Q) and increasing concentrations (from 0.5 to 3 μg) of wild-type CIC (left panel) or increasing concentrations of mutant CIC/W37A (right panel). Membranes were probed with F11G3 to determine the levels of oligomers. Almost no oligomers were detected in any of the measured conditions. * indicated change in exposure of the membrane. (**D**) ELISA using F11G3 shows that cells transfected with ATXN1(82Q) have high levels of oligomers, but cells transfected with ATXN1(30Q) had almost no detectable oligomers. Increasing amounts of oligomers were observed when cells transfected with ATXN1(82Q) were co-transfected with increasing amounts of wild type CIC but not mutant CIC/W37A. n = 4. *p < 0.05, **p < 0.01 and ***p < 0.001.

The following figure supplement is available for figure 5:

**Figure supplement 1**. CIC facilitates ATXN1 oligomerization.

transfected with nonpathogenic ATXN1(30Q) failed to produce oligomers, whether they were co-transfected with CIC, CIC/W37A, or not co-transfected at all (*Figure 5D*).

We also transiently transfected mRFP-ATXN1(82Q) stable cells with the N-terminal region of CIC, which contains the ATXN1 binding region (*Lam et al., 2006*). As a negative control, we transfected N-terminal CIC with the W37A mutation. Cells transfected with N-terminal wild-type CIC developed many more oligomeric inclusions than those transfected with mutant CIC or controls (*Figure 5—figure supplement 1*). This effect was specific to oligomeric inclusions.

We next studied the effect of CIC on the aggregation kinetics of ATXN1 in vitro. Because full-length ATXN1 is insoluble and difficult work with in vitro (*de Chiara et al., 2005*; *Lam et al., 2006*; *Goldschmidt et al., 2010*), we used ATXN1's AXH domain, which is the region that interacts with CIC and is also responsible for a large part of the protein's toxicity in mice and fruit flies (*Tsuda et al., 2005*) and its tendency to aggregate (*Figure 6—figure supplement 1*). When the AXH peptide alone was stirred at 37°C, soluble oligomers formed in as little as 12 hr but were no longer detectable after 48 hr. When AXH was mixed with CIC in a 1:1 molar ratio, however, oligomeric species were more abundant and persisted much longer, being detectable for up to 72 hr (*Figure 6A*). When we blocked AXH-CIC binding (using CIC W37A), AXH oligomer kinetics resembled those of AXH alone (*Figure 6A*).

To better characterize the effect of CIC on the aggregation kinetics of ATXN1, we utilized the conformational antibody OC that specifically detects fibrillar oligomers and fibrils (*Kayed et al., 2007*). It has been shown that these fibrillar oligomers are more mature oligomers, more compact and less toxic than the soluble ones detected with A-11 and F11G3, known as prefibrillar oligomers (*Krishnan et al., 2012*; *Guerrero-Munoz et al., 2014b*). Despite the structural similarity of fibrillar oligomers and fibrils, the former are not positive for Thioflavine (*Glabe, 2008*). As expected, AXH either alone or with mutant CIC W37A formed significantly more fibrillar oligomers at 24 hr than it did with wild-type CIC (*Figure 6B*). This result suggests that CIC binding to ATXN1 promotes the formation of and stabilizes soluble prefibrillar oligomers and tends to divert ATXN1 from forming more mature and less toxic aggregates such as fibrillar oligomers and fibrils. To solidify this interpretation, we performed a Thioflavin T (Thio-T) assay to determine the effect of CIC on the formation of AXH fibrils. Fluorescence measurements of Thio-T showed that wild-type CIC diminished the formation of AXH fibrils (*Figure 6C*). The results obtained from our aggregation kinetics assay suggested that CIC not only promotes AXH oligomerization but also stabilizes the oligomers, retarding the kinetics of AXH aggregation to fibrils.

Having demonstrated that increasing levels of CIC stabilize ATXN1 polyQ oligomers in cells (*Figure 5*), we sought to determine whether the particular susceptibility of the cerebellum to ATXN1 toxicity could be attributable, at least in part, to CIC levels in the tissue. We measured the levels of ATXN1 and CIC in the cerebellum and cortex of wild type mice and found that, under these physiological conditions, CIC levels are higher in the cerebellum than in the cortex, but ATXN1 levels are lower (*Figure 7*). The amount of CIC available relative to ATXN1 is thus considerably higher in the cerebellum than in the cortex. This result suggests that in brain regions containing high levels of CIC in relation to ATXN1 (such as the cerebellum), CIC stabilizes ATXN1 in its oligomeric conformation, but where the ratio of CIC to ATXN1 is lower, ATXN1 is not stabilized in its oligomeric confirmation and continues its process of forming non-toxic NIs.

## Discussion

An overwhelming number of studies have shown that structurally related oligomers are toxic (*Chiti and Dobson, 2006*) and that the soluble rather than the fibrillar mutant protein is associated with a toxic gain of function (*Saunders and Bottomley, 2009*; *Zoghbi and Orr, 2009*). Oligomers have been specifically associated with Alzheimer disease, Parkinson disease, and frontotemporal dementia, and it has been proposed that the disease pathology could be caused solely by soluble toxic entities of Aβ, α-synuclein or tau, respectively (*Lesne et al., 2006*; *Berger et al., 2007*; *Shankar et al., 2008*). Yet the molecular mechanism(s) of this toxicity have remained poorly understood. Not since Prusiner proposed the existence of a 'protein X' that might help form the toxic species has attention been paid to a possible requirement for interaction with another protein partner to form toxic oligomers (*Telling et al., 1995*). Here we have demonstrated not only that polyQ ATXN1 adopts an oligomeric conformation and that this conformation is pathogenic in cellular and animal models of SCA1, but that the native binding

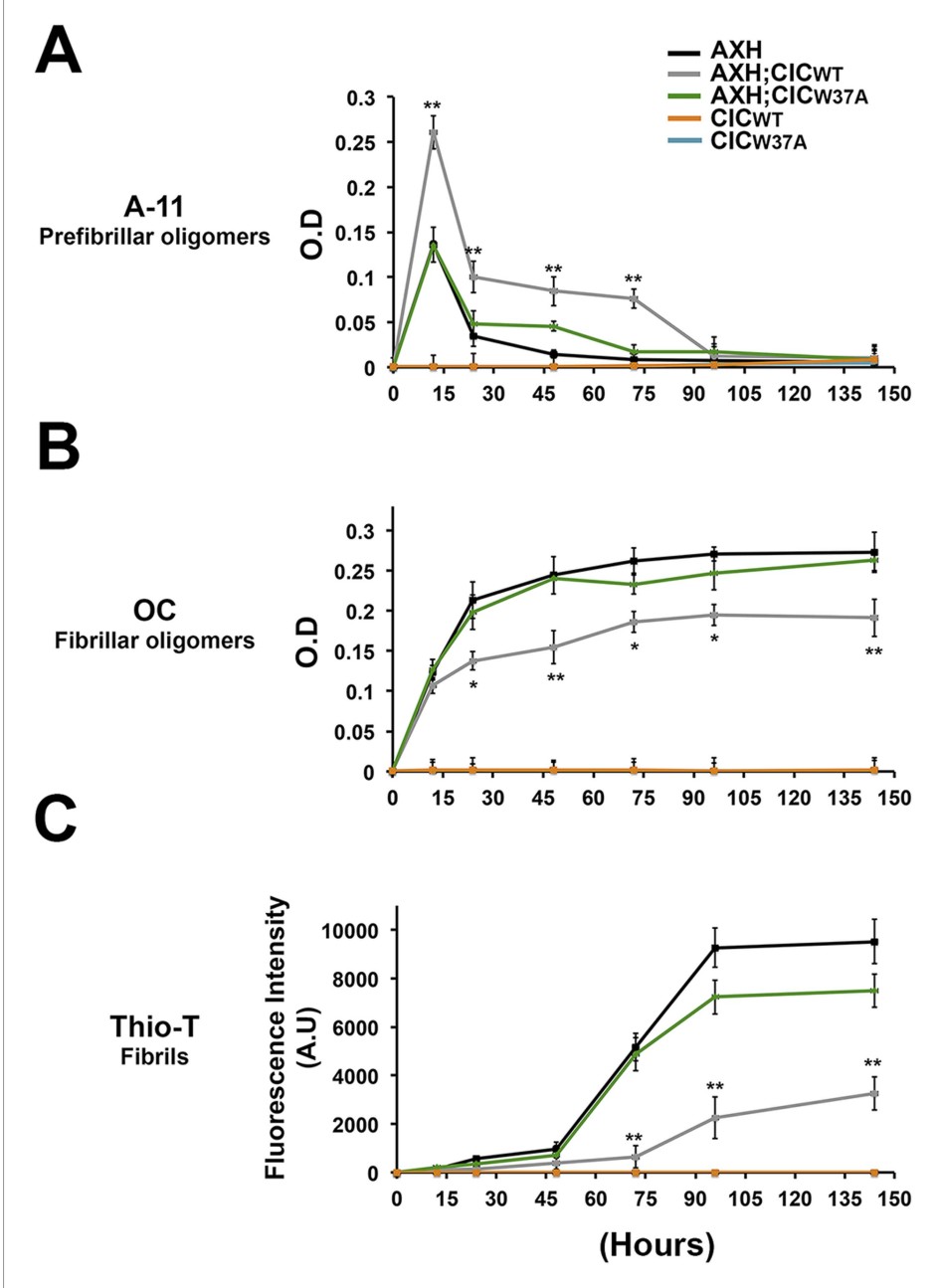

**Figure 6**. CIC stabilizes ATXN1 oligomers. (**A**) ELISA using anti-prefibrillar oligomer antibody A-11 shows that the N-terminal fragment of wild-type CIC stabilized AXH fragments in the oligomeric conformation, unlike the N-terminal fragment of mutant CIC W37A or AXH incubated alone. Wild-type and mutant CIC alone did not oligomerize at any time point. (**B**) ELISA using anti-fibrillar oligomer antibody OC shows that only wild type, not mutant, CIC partially inhibits the formation of fibrillar oligomers. (**C**) Thioflavin T assay demonstrated that wild-type but not mutant CIC blocks the formation of AXH fibrils. *p < 0.05, **p < 0.01.

The following figure supplement is available for figure 6:

**Figure supplement 1**. The AXH domain of ATXN1 has a high energy for aggregation.

partner of ATXN1, CIC, is essential for the stabilization of oligomeric polyQ ATXN1. We believe the data provide compelling reason to investigate the role of native partners involved in the toxicity of protein oligomers in other neurodegenerative proteinopathies.

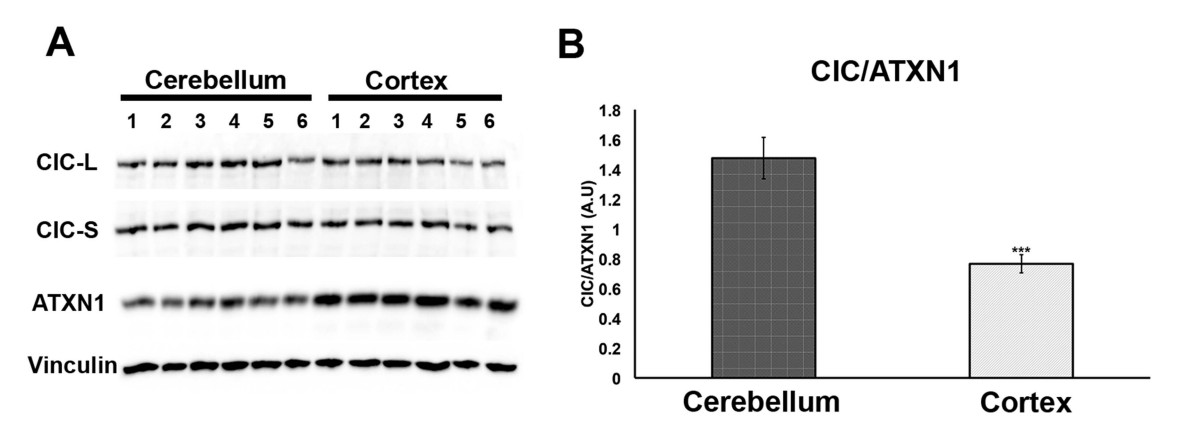

**Figure 7**. Comparison of CIC and ATXN1 protein levels in cerebellum and cortex. CIC levels are higher in the cerebellum than in the cortex, while ATXN1 levels are lower in the cerebellum than in the cortex, making the amount of CIC per ATXN1 much higher in the cerebellum.

Our study also provide an answer to the longstanding conundrum of differential regional vulnerability, i.e., why a ubiquitiously expressed protein should prove toxic to only certain brain regions and leave others relatively unharmed. It has been particularly perplexing that regions that develop insoluble nuclear inclusions, which were long thought to be toxic, tend to escape SCA1 pathology (*Watase et al., 2002*). We had previously shown that polyQ ATXN1 exerts toxicity through its native complexes containing CIC rather than through aberrant interactions with novel proteins (*Lam et al., 2006*) and that reducing CIC levels by 50% rescues the phenotype of a mouse model for SCA1, improving motor functions and increasing survival (*Fryer et al., 2011*). The data from the current study provide insight into how CIC modulates pathogenesis and why it is particularly toxic in the cerebellum: CIC stabilizes toxic oligomers and exists in a particularly high ratio with respect to ATXN1 in cerebellar neurons. In the SCA1 knockin mice, the abundance of these oligomers correlated directly with behavioral deficits; without CIC, in vitro and in vivo, ATXN1 oligomers continue on the aggregation pathway and co-assemble into less-toxic fibrillar oligomers, amorphous aggregates and/or fibrils (*Figure 8*). SCA1 is therefore triggered not just by a conformational change in ATXN1 but by interactions with a specific partner in a specific brain region that brings about oligomerization and stabilization. The partner not only has to be present, but its levels have to be high enough to stabilize a considerable proportion of ATXN1 oligomers in order to drive pathogenesis. These observations do much to explain why pathology is relatively localized in SCA1 despite the ubiquitous expression of ATXN1. It seems likely that the physiological cellular environment will prove to be the determining factor in promoting and stabilizing the respective pathological proteins in their toxic conformations in other polyglutamine diseases and proteinopathies.

Amyloid oligomers are characterized by hydrophobic domains; ATXN1 dimers form a hetero-tetramer with CIC dimers by hydrophobic interactions (*Bemporad and Chiti, 2012*; *Kim et al., 2013*). In light of this, we propose that CIC interacts with oligomeric polyglutamine-expanded ATXN1 dimers through the exposed hydrophobic domains, which in turn stabilizes the complexes. In other words, CIC—a native ATXN1 interactor—does not discern between a nonpathogenic wild type ATXN1 dimer and a pathologically expanded ATXN1 dimer, and interacts with either; it simply binds to ATXN1 to form a functional transcriptional repressor complex or a dysfunctional oligomeric complex, depending on the polyglutamine tract. The length of the polyQ tract is essential to the accumulation, aggregation and oligomerization of ATXN1, as has been postulated for other polyglutamine diseases (*Legleiter et al., 2010*). Given that CIC interacts with ATXN1 regardless of polyQ expansion (*Lam et al., 2006*) and that CIC stabilizes AXH oligomers, we believe that CIC stabilizes ATXN1 oligomers independent of the polyQ tract length. Nevertheless, a polyglutamine expansion is necessary to initiate the oligomerization process under physiological conditions, in the context of the full-length protein. To our knowledge, this is the first evidence that a native interactor of a toxic protein can stabilize its oligomeric form to promote its bioavailability and downstream toxicity. Furthermore, ATXN1

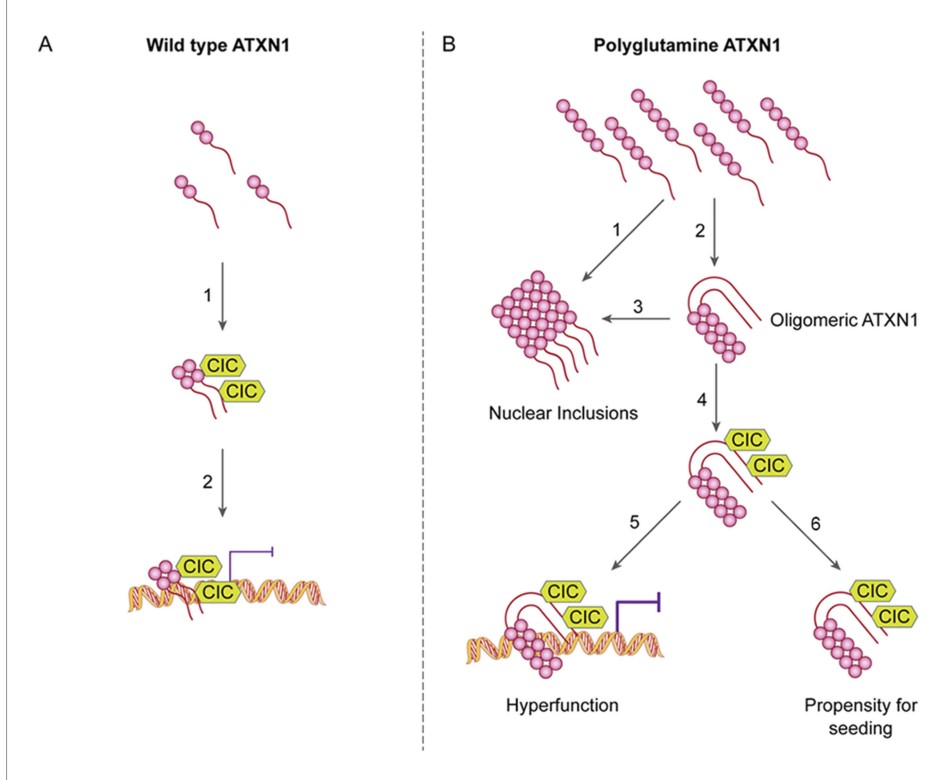

**Figure 8**. ATAXIN-1 oligomer complexes with seeding capability correlate with disease progression in spinocerebellar ataxia type 1: a model. Under normal conditions (**A**, left panel), wild type ATXN1 forms a transcriptional repressor complex with CIC (1) and binds to DNA (2). Under pathological conditions (**B**, right panel), polyQ-expanded ATXN1 accumulates and can directly form nuclear inclusions (1) or it can adopt an oligomeric conformation (2). These oligomers form nuclear inclusions (3) or form a stable oligomeric complex with CIC (4) that could: act as a dysfunctional transcriptional repressor complex (5) or be released into the extracellular space and be internalized by neighboring neurons, seeding the formation of new oligomers (6). The toxic effect of oligomeric complexes might be mitigated by reducing the total levels of polyQ ATXN1 or hindering the propagation of these toxic complexes (e.g., by immunotherapy).

oligomeric complexes were able to penetrate cells and seed the formation of new ATXN1 oligomers without forming fibrillar inclusions. This could be explained by the addition of monomers to the ends of oligomers, which is kinetically more favorable than the assembly of oligomers into fibrils via sheet stacking interactions (*Wu et al., 2010*).

Our findings for SCA1 are likely to be applicable to other members of the larger family of neurodegenerative proteinopathies. It should be worthwhile, for example, to explore whether native interactors of tau and α-synuclein might promote or hinder their formation of toxic oligomers. Without ruling out the role of non-native interactors in protein aggregation, we propose that neuropathological features of a specific disease are determined by both transient and stable native protein interactions and that these interactions in turn affect the amyloidogenic properties of the disease-associated protein. Thus, blocking certain interactions that mediate the formation or stabilization of oligomeric toxic entities, or modulating the downstream effects of such interactions, could be an attractive treatment option for this class of diseases.

## Materials and methods

### Mouse models and preparation of brain extracts
All mouse procedures were approved by the Institutional Animal Care and Use Committee for Baylor College of Medicine and Affiliates. *Atxn1*$^{154Q/+}$, and *Atxn1*$^{-/-}$ mice have been previously described (*Lorenzetti et al., 2000*; *Watase et al., 2002*) and were backcrossed to C57BL/6 for more than ten

generations. The $Atxn1^{154Q/+};Cic^{L+/-}$ mouse was previously generated and characterized (*Fryer et al., 2011*). Briefly, male $Atxn1^{154Q/+}$ animals were crossed with female $Cic^{L+/-}$ to obtain the following genotypes: wild type, $Cic^{L+/-}$, $Atxn1^{154Q/+}$, and $Atxn1^{154Q/+};Cic^{L+/-}$. Twenty-four weeks-old (unless stated otherwise) mouse cerebella were dissected and then lysed in 0.5% Triton buffer (0.5% triton X-100, 50 mM Tris pH 8, 75 mM NaCl) supplemented with protease and phosphatase inhibitors (Sigma, St. Louis, MO). The protein lysate was then incubated on ice for 20 min and centrifuged at 13,200 r.p.m. for 10 min at 4°C, and the supernatants were portioned into aliquots, snap-frozen, and stored at −80°C until used.

## Column fractionation
Size exclusion chromatography (SEC) was performed as previously described (*Park et al., 2013*). Briefly, 700 µl of triton soluble sample was applied to a Superose 6 GL 300 column (GE Healthcare and Life Science, United Kingdom) and run at 0.3 ml/min using a 0.1% triton column buffer. Fractions were collected every 1 ml.

## Cell-based seeding assay
A stable Daoy mRFP-ATXN1(82Q) cell line was generated as previously described (*Park et al., 2013*). Cells were plate in 24 well plates ($2*10^4$ cells/ml). 24 hr later, cells were treated with SEC fractions (1.5 µg of total protein) from mouse cerebellum. After 10 hr of treatment, cells were fixed with methanol at −80°C for 45 min. RFP-positive inclusions ranging from 350 to 900 nm were considered oligomeric; inclusions larger than 900 nm and not detected by F11G3 were considered fibrillar. One hundred cells were quantified per group in triplicates. Analyses were manually performed with Image J. For blocking assays, each fraction was mixed with each antibody (2.5 mg/ml) in a ratio 1:1 (vol/vol) for 1 hr and then added to the cells.

## Rotarod assay
Motor coordination was assessed on the Rotarod assay as previously described (*Park et al., 2013*), with four trials for 4 days using 8-, 18- and 28-week-old mice for the correlation measurements. The tester was blinded to animal genotype and treatment.

## Western blots analysis
Triton-soluble fractions of brain extracts were run on Tris-Glycine 5% gels, in nonreducing conditions to avoid altering the oligomeric conformation (NuPage sample buffer without β-mercaptoethanol) and subsequently transferred onto nitrocellulose. After blocking overnight at 4°C with 10% nonfat dried milk, membranes were probed for 1 hr at room temperature with anti-oligomer antibody F11G3, provided by Dr Rakez Kayed (1:1000), anti-ATXN1 antibody 11750 (1:3000), anti-ATXN1 antibody 11NQ, or anti-CIC antibody (1:4000) diluted in 5% nonfat dried milk. 11750 and 11NQ immunoreactivity was detected with horseradish peroxidase conjugated anti-rabbit IgG (1:8000; Jackson ImmunoResearch Laboratories, West Grove, PA); anti-mouse IgM (1:6000; Jackson ImmunoResearch Laboratories) was used for F11G3 and anti-guinea pig IgG (1:8000; Jackson ImmunoResearch Laboratories). For signal detection, ECL Plus (Amersham-Pharmacia, Piscataway, NJ, USA) was used.

## Brain section IF
Paraffin sections were deparaffinized, rehydrated, and washed in 0.01 M PBS 3 times for 5 min each time. After blocking in normal goat serum for 1 hr, sections were incubated overnight with rabbit anti-ATXN1 antibody 11750 (1:700). The next day, the sections were washed in PBS 3 times for 10 min each time and then incubated with goat anti-rabbit IgG Alexa Fluor 568 (1:700; Invitrogen, Grand Island, NY) for 1 hr. The sections were then washed 3 times for 10 min each time in PBS before incubation overnight with mouse anti-oligomers F11G3 (1:300). The next day, the sections were washed in PBS 3 times for 10 min each before incubation with goat anti-IgM Alexa Fluor 488 (1: 700; Invitrogen) for 1 hr. Sections were washed and mounted in Vectashield mounting medium with DAPI (Vector Laboratories, Burlingame, CA). The sections were examined using a Zeiss LSM 710 confocal microscope.

## Immunohistochemistry
Immunohistochemistry was performed on paraffin-embedded sections. In brief, sections (5 µm) were deparaffinized and rehydrated. Primary antibodies were detected with biotinylated goat anti-mouse

IgG (1:2000; Jackson ImmunoResearch Laboratories), biotinylated goat anti-mouse IgM (1:1500), or biotinylated goat anti-rabbit IgG (1:1800) (all from Jackson ImmunoResearch Laboratories) and visualized using an ABC reagent kit (Vector Laboratories), according to the manufacturer's recommendations. Bright-field images were acquired using a Carl Zeiss Axio Imager M2 microscope, equipped with an Axio Cam MRc5 color camera (Carl Zeiss, Germany). Sections were counterstained with hematoxylin (Vector Laboratories) for nuclear staining. The following antibodies were used for immunostaining: rabbit anti-ATXN1 antibody 11750 (1:700), rabbit anti-oligomer antibody A-11 (1:600), mouse anti-oligomer antibody F11G3 (1:100), and mouse anti-calbindin antibody (1:450).

## Cell toxicity assays

Cerebellar granular precursor cells were obtained from E16 wild-type C57BL6 mice. Cells were maintained in DMEM (Life Technologies, Inc., Invitrogen, Carlsbad, CA, USA) supplemented with 10% FBS, glutamine (4 mM), penicillin (200 U/ml), streptomycin (200 µg/ml), and sodium pyruvate (1 µM). Cells were maintained at 37°C in 5% $CO_2$. Cells (~10,000/well) were plated in 96-well plates (Corning Glassworks, Corning, NY) and grown overnight. SEC fractions (0.3 µg of total protein) from mouse cerebellum were added to the cells and incubated for 24 hr. Cell viability was assessed spectrophotometrically using a 3-[4,5-dimethylthiazol-2-yl]-2,5-diphenyltetrazolium bromide (MTT)-based assay according to the manufacturer's specifications (Sigma–Aldrich, St. Louis, MO, USA).

## Immunocytochemistry (ICC)

CGP or stable Daoy mRFP-ATXN1(82Q) cells on the coverslip were fixed and permeabilized after their specific treatment. The samples were blocked for 1 hr in 5% goat serum and incubated overnight at 4°C with anti-Nuj1 for CPG cells (1:150, Covance, Cranford, NJ). For the Daoy mRFP-ATXN1(82Q) cells, samples were incubated overnight with anti-oligomer F11G3 (1:100), anti-ATXN1 11,750 (1:500) or anti-Myc (1:500, GenScript). In all cases cells were then washed and incubated with an Alexa 488-conjugated goat anti-mouse or anti-rabbit antibody (1:700, Invitrogen) for 1 hr at room temperature. DAPI was used for nuclear staining (1:4000, Invitrogen). Cells were then washed for 30 min and cover slipped.

## ELISA

For ELISA, plates were coated with 10 µl of sample using 0.1 M sodium bicarbonate (pH 9.6) as a coating buffer, followed by incubation for 1 hr at 37°C, washing 3 times with Tris-buffered saline with very low (0.01%) Tween 20 (TBS-T), and then blocking for 1 hr at 37°C with 10% BSA. The plates were then washed three times with TBS-T; F11G3 (1:500), A-11 (1:1000), 11,750 (1:2000), OC (1:1000) or Tubulin (1:2000) antibodies (diluted in 5% nonfat milk in TBS-T) were added and allowed to react for 1 hr at 37°C. The plates were then washed 3 times with TBS-T, and 100 µl of horseradish peroxidase-conjugated anti-mouse IgM, anti-mouse IgG or anti-rabbit IgG (diluted 1:10,000 in 5% nonfat milk in TBS-T; Promega, Madison, WI, USA) were added, followed by incubation for 1 hr at 37°C. Finally, plates were washed 3 times with TBS-T and developed with 3,3,5,5-tetramethylbenzidine (TMB-1 component substrate) from KPL (Gaithersburg, MD, USA). The reaction was stopped with 100 µl of 1 M HCl, and samples were read at 450 nm.

## Protein expression and purification

The AXH domain was expressed and purified as previously described (*Kim et al., 2013*). Briefly, the AXH domain (amino acids 563–689) from human ATXN1 was cloned into a pET28a vector, generating an N-terminal His tag. The AXH domain was expressed in *Escherichia coli* BL21(DE3) and purified with an Ni-affinity column. The N-terminal His tag was cleaved by TEV protease (Life Technologies) and further purified with a Hitrap Q (GE Healthcare) anion exchange chromatography and with Superdex 75 (GE Healthcare) size exclusion chromatography (SEC). The CIC N-terminal fragment (amino acids 1–117), wild-type or mutant (W37A), was cloned into a pGEX-4T-1 vector, generating a GST tag. The N-terminal CIC was expressed in *Escherichia coli* BL21(DE3) and purified in a GST-affinity column.

## AXH oligomer formation

After AXH purification, the buffer was exchanged by dialysis to 20 mM Tris–HCl (pH 7.0), with 20 mM NaCl, 2 mM β-mercaphtoethanol and 0.01% (wt/vol) $NaN_3$ and then concentrated to a final

concentration of 0.3 mg/ml. The AXH solution was stirred at 37°C for 7 days. Aliquots were collected at different time points and stored at −80°C. To determine the effect of CIC in AXH oligomer formation, both peptides were mixed in a 1:1 molar ratio and stirred at 37°C for 7 days. Aliquots were also collected at different time points and stored at −80°C. All experiments were performed in triplicate.

## AXH oligomer quantification

For AXH oligomers quantification, ELISA was performed as described above. Briefly, 10 µl of sample was coated overnight. The anti-oligomer antibody A-11 (1:1000) was utilized to detect oligomers and the anti-ATXN1 antibody 11750 (1:500) was used to quantify total AXH.

## Thioflavin T assay

Thioflavin T (Sigma) binding was measured using a POLARstar OMEGA plate reader (BMG Labtechnologies, Melbourne, VIC, Australia) with 440-10 nm/520 nm excitation/emission filters set. 1 µl of 0.3 mg/ml of protein sample were mixed with 250 µl of 5 µM ThT, 50 mM glycine buffer (pH 8.5). Fluorescence intensity values of samples were obtained by subtraction of blank.

## Hydrophobic interaction column

One ml of the sample was diluted 20 times in 1.7 M Ammonium Sulfate buffer and then loaded on to the Hydrophobic interaction column column (1 ml Phenyl Sepharose Fast Flow, low substitution, from GE) at 1 ml/min. Fractions were collected every 0.5 ml. An AKTA UPC 10 FPLC system from GE at 4° was used for all chromatography steps.

## Analysis of ATXN1 pathology

To determine the percentage of PCs with ATXN1 oligomers, 5 µm brain sections were stained with anti-calbindin antibody as described above to quantify the total number of PCs in the cerebellum. The number of PCs positive for calbindin was considered as 100%. The adjacent sections were immunostained for oligomers using F11G3 or A-11. For the quantification of nuclear inclusions, we stained ATXN1 using 11750 antibody and performed nuclear staining with hematoxylin. We considered the total amount of nucleus in the cortex as 100% of the nucleus. Data were analyzed using post-hoc test.

## Antibody blocking experiment

The antibody F11G3 was preincubated for 40 min with Aβ oligomers, IAPP oligomer, Aβ monomers or IAPP monomers. After the incubation period, this antibody was used to perform double IF experiments with 11,750 in $Atxn1^{154Q/+}$ mouse cerebellar sections.

## Oligomers stability assay

SEC fractions from $Atxn1^{154Q/+}$ mouse cerebella were incubated up to 48 hr at 37°C. The presence of oligomers in each fraction at each time point was determined by dot-blot using F11G3 (1:1000).

## Filter retardation assay

A total of 50 µg of SEC fractions protein in 200 ml of 2% SDS was boiled for 5 min and run through a dot blot apparatus under a vacuum onto a cellulose acetate membrane. Membrane was then washed 3x with 0.1% SDS and then blocked in 5% milk and subject to western blot analysis using F11G3 (1:1000), OC (1:1000) and 11750 (1:8000).

## Cell transfection

Mammalian expression vectors Myc-CiC/WT, Myc-CiC/W37A, GST-ATXN1(82Q) and GST-ATXN1 (30Q) have been described previously (*Lim et al., 2008*; *Kim et al., 2013*). Hela cells were cultured in DMEM medium with 10% fetal bovine serum and were plated in six-well plates 1 day before transfection. On the day of transfection, corresponding DNA constructs were transfected with Lipofectamine 2000 (Invitrogen). In all cases 100 ng of GST-ATXN1(30Q) or GST-ATXN1(82Q) were used for transfection. For the CIC construct we transfected 0.5, 1, 2 or 3 µg. Two days later, cells were lysed for 20 min at 4°C in lysis buffer (150 mM NaCl, 50 mM Tris–HCl at pH 7.5, 1 mM EDTA, 0.1% Triton X-100, 0.1% Tween-20, complete protease inhibitor cocktail [Roche,

Switzerland], PhosSTOP phosphatase inhibitor cocktail [Roche]). After centrifugation, soluble fraction was used for western blot and ELISA measurements.

## Immunoprecipitation

Lysis was performed on ice for 20 min with brief vortexing using in 0.5% Triton buffer (0.5% triton X-100, 50 mM Tris pH 8, 75 mM Nacl) supplemented with protease and phosphatase inhibitors (Sigma). Cell debris were removed by centrifugation (20 min, 15,000 r.p.m, 4°C) and pre-cleared with un-conjugated beads. In parallel, 2 μg of antibody was conjugated to Dynabeads for 1 hr at 4°C with rocking. Lysate was then added to the conjugated beads overnight at 4°C. Beads were then washed 5 × 500 μl of lysis buffer before being eluted using sample buffer and boiling for 10 min.

## Statistical analysis

Data are represented as mean ± SEM. Different conditions were compared using one- or two-way ANOVA followed by the indicated *post hoc* test to compare controls with treatment groups and/or genotypes or by two-tailed Student's test, as appropriate.

## Acknowledgements

We thank the members of the Zoghbi, Orr and Kayed laboratories for suggestions and discussions, and V Brandt for critical reading of the manuscript. We are grateful to Luis Vilanova for excellent technical assistance. This work was supported by a Howard Hughes Medical Institute Collaborative Innovation Awards grant, the Robert A and Renee E Belfer Family Foundation and grants NIH/NINDS R01 NS027699-17, 3R01 NS027699-25S1. MWCR wants to thank The Canadian Institutes of Health Research Fellowship (201210MFE-290072-173743). We thank MV Bhaskar and the art team of TNQ for drawing *Figure 8*. We also appreciate the assistance of the confocal microscopy and mouse behavioral cores of the Baylor College of Medicine (BCM) Intellectual and Developmental Disabilities Research Center (1U54 HD083092).

# Additional information

## Competing interests

HYZ: Senior editor, *eLife*. The other authors declare that no competing interests exist.

## Funding

| Funder | Grant reference | Author |
|---|---|---|
| Howard Hughes Medical Institute (HHMI) | | Huda Y Zoghbi |
| Robert A and Renee E Belfer Family Foundation | | Cristian A Lasagna-Reeves, Maxime WC Rousseaux, Paymaan Jafar-Nejad, Nan Lu, Huda Y Zoghbi |
| National Institute of Neurological Disorders and Stroke (NINDS) | 3R01 NS027699-25S1 | Cristian A Lasagna-Reeves |
| National Institute of Neurological Disorders and Stroke (NINDS) | R01 NS027699-17 | Huda Y Zoghbi |

The funders had no role in study design, data collection and interpretation, or the decision to submit the work for publication.

## Author contributions

CAL-R, MWCR, RR, NL, Conception and design, Acquisition of data, Analysis and interpretation of data, Drafting or revising the article; MJG-M, JP, US, AL, Conception and design, Acquisition of data, Analysis and interpretation of data; PJ-N, Acquisition of data, Drafting or revising the article, Contributed unpublished essential data or reagents; HTO, Analysis and interpretation of data, Drafting or revising the article; RK, Analysis and interpretation of data, Drafting or revising the

article, Contributed unpublished essential data or reagents; HYZ, Conception and design, Analysis and interpretation of data, Drafting or revising the article

## Ethics

Animal experimentation: This study was performed in strict accordance with the recommendations in the Guide for the Care and Use of Laboratory Animals of the National Institutes of Health. All of the animals were handled according to approved institutional animal care and use committee (IACUC) protocols (#AN-1013) of Baylor College of Medicine

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
