## [Decision Letter]

Thank you for sending your work entitled "A native interactor scaffolds and stabilizes toxic Ataxin-1 oligomers in SCA1" for consideration at *eLife*. Your article has been favorably evaluated by a Senior editor, a member of our Board of Reviewing Editors, and three reviewers. We add the individual reports which are all three very positive. No further experimental work seems needed and careful consideration of the different suggestions could help to improve the final version of the work.

Reviewer #1:

I like this story about the toxicity of prefibrillar ATXN1 oligomers a lot. I think it convincingly shows, primarily using conformation specific antibodies, that the fraction of the polyQ expanded mutant protein that forms prefibrillar toxic oligomers depends on the level of a known interactor of native ATXN1. This model makes much sense for polyQ proteins since the expansion is likely to leave other domains natively folded—the generality is of course unclear at this moment. The observations are important because they provide a way to understand the otherwise perplexing stereospecificity of the cytotoxcicity of aggregating proteins, which is one of the main outstanding questions in the field: why is the same aggregating protein toxic to some cells but not to others.

The work is well done and the manuscript is well written. My main concern regards the conformational antibody itself. I think the authors should discuss in the manuscript how F11G3 differs from the well-known A11. There are serious concerns about the variations in the properties of this antibody both batch to batch and between suppliers. Is F11G3 better behaved, or should readers worry about the same flaws (which come down to reproducibility issues)? I think in terms of the credibility of the story I would provide some more support for the robustness of this (to me) relatively new antibody.

Reviewer #2:

This is a very exciting and timely manuscript that both nicely builds on the authors’ previous work and provides important new findings relating to protein propagation in CAG repeat diseases. The use of the antibody developed by Dr. Kayed is a strength. This has been a very active area of research over the last several years, having been shown for FTD, PD and other disorders and more recently in cells for HD. This is a very elegant study showing that Capicua, which was shown as critical to SCA1 previously by Zoghbi and Orr, is linked to formation of oligomers and seeding of aggregation. They convincingly show that oligomers correlate with disease progression in vivo in SCA1 transgenic mice using a multitude of assays. Any of these in isolation would be somewhat correlative, but taken together they build an extremely strong case. The data also provide support for the hypothesis that the protein interactions that drive the formation of these toxic oligomeric structures may explain some of the selectivity that drives disease in Purkinje cells.

Reviewer #3:

This manuscript is a thorough and wide-ranging study of the role of oligomers in SCA1, one of nine polyglutamine neurodegenerative disorders. The authors marshal an impressive range of methods and employ multiple mouse and cell models to ascertain the presence of disease protein oligomers in susceptible brain tissue/cells and explore interactions between such oligomers and a well-established disease protein interacting partner, Capicua (CIC1). There are many strengths to the paper. It serves as an important addition to the field of polyglutamine diseases specifically, and to neurodegenerative proteinopathies more generally. I have several small points of clarification and questions for the authors, but no truly major concerns.

Two points of moderate significance:

1) The distinction of fibrillar from oligomeric inclusions, which is based in part on size, is not particularly well supported in the paper. It is a critical distinction for many of the authors' result and ultimate conclusion, and may benefit from additional description.

2) The tight correlation between degree of motor dysfunction in SCA1 knock-in mice and levels of ATXN1 oligomers, while striking and intriguing, is likely heavily weighted toward the young mice and old mice which respectively show low and high levels of oligomers on average (these young and old mice presumably differ in many other respects as well). Ideally, though not essential for this paper, the authors would analyze a set of mice of the same, intermediate age with varying degree of motor dysfunction and determine whether, at this intermediate age, levels of oligomers indeed do correlate closely with degree of motor dysfunction.

---

## [Author Response]

Reviewer #1:

*The work is well done and the manuscript is well written. My main concern regards the conformational antibody itself. I think the authors should discuss in the manuscript how F11G3 differs from the well-known A11. There are serious concerns about the variations in the properties of this antibody both batch to batch and between suppliers. Is F11G3 better behaved, or should readers worry about the same flaws (which come down to reproducibility issues)? I think in terms of the credibility of the story I would provide some more support for the robustness of this (to me) relatively new antibody*.

We greatly appreciate the reviewer’s comments. Regarding the question about differences between F11G3 and A-11, because we focused on use of F11G3 it is hard for us to make claims regarding how the two compare. F11G3 has been previously extensively characterized and compared with A-11 (Guerrero-Munoz et al. 2014. Neurobiology of disease 71, 14-23).

To make things clearer we now state this in the Results section and we also indicate in the Results and in the Materials and methods sections that F11G3 is a mouse monoclonal antibody that can be provided by Dr. Kayed and is not commercially available in contrast to A-11 that is a rabbit polyclonal that have been sold commercially.

Reviewer #3:

*1) The distinction of fibrillar from oligomeric inclusions, which is based in part on size, is not particularly well supported in the paper. It is a critical distinction for many of the authors' result and ultimate conclusion, and may benefit from additional description*.

The distinction between different inclusions was based not only on the size but also on F11G3 reactivity. Nevertheless, we agree that additional description would be helpful. Therefore in the subsection “ATXN1 oligomers seed the formation of endogenous ATXN1 oligomers in cell culture”, we stated: “larger inclusions are more complex and composed of higher order aggregated, likely to be fibrillar in nature”.

*2) The tight correlation between degree of motor dysfunction in SCA1 knock-in mice and levels of ATXN1 oligomers, while striking and intriguing, is likely heavily weighted toward the young mice and old mice which respectively show low and high levels of oligomers on average (these young and old mice presumably differ in many other respects as well). Ideally, though not essential for this paper, the authors would analyze a set of mice of the same, intermediate age with varying degree of motor dysfunction and determine whether, at this intermediate age, levels of oligomers indeed do correlate closely with degree of motor dysfunction*.

In the current manuscript we already analyzed a set of mice of intermediate age (18 weeks old) with varying degree of motor dysfunction (young would be 5-8 weeks and old would be 35-50 weeks). Please refer to Figure 2 where we observed the correlation between the levels of oligomers and the degree of motor dysfunction in these mice.